# Mapping influenza transmission in the ferret model to transmission in humans

**Michael G Buhnerkempe[1,2]\*[†], Katelyn Gostic[1][†], Miran Park[1], Prianna Ahsan[1], Jessica A Belser[3], James O Lloyd-Smith[1,2]**

[1]Department of Ecology and Evolutionary Biology, University of California, Los Angeles, Los Angeles, United States; [2]Fogarty International Center, National Institutes of Health, Bethesda, United States; [3]Influenza Division, National Center for Immunization and Respiratory Diseases, Centers for Disease Control and Prevention, Atlanta, United States

**Abstract** The controversy surrounding 'gain-of-function' experiments on high-consequence avian influenza viruses has highlighted the role of ferret transmission experiments in studying the transmission potential of novel influenza strains. However, the mapping between influenza transmission in ferrets and in humans is unsubstantiated. We address this gap by compiling and analyzing 240 estimates of influenza transmission in ferrets and humans. We demonstrate that estimates of ferret secondary attack rate (SAR) explain 66% of the variation in human SAR estimates at the subtype level. Further analysis shows that ferret transmission experiments have potential to identify influenza viruses of concern for epidemic spread in humans, though small sample sizes and biological uncertainties prevent definitive classification of human transmissibility. Thus, ferret transmission experiments provide valid predictions of pandemic potential of novel influenza strains, though results should continue to be corroborated by targeted virological and epidemiological research.

\*For correspondence: michael. buhnerkempe@ucla.edu

[†]These authors contributed equally to this work

**Competing interests:** The authors declare that no competing interests exist.

## Introduction

The emergence of deadly animal-origin influenza viruses in human populations, such as influenza A(H5N1) (*Chan, 2002*; *Li et al., 2004*), and influenza A(H7N9) (*Gao et al., 2013*; *Gong et al., 2014*; *Li et al., 2014*), has underscored the need to rapidly determine the pandemic potential of novel strains found in humans or in zoonotic reservoirs. Although characterizing human transmissibility of emerging influenza viruses is a perpetual challenge, animal models are often used to characterize transmission among mammals, which can be viewed implicitly as a preliminary screen for pandemic potential in humans. Ferrets are the preferred animal model for influenza transmission studies because clinical signs, pathogenesis and sialic acid distribution are similar in ferrets and humans (*Maher and DeStefano, 2004*; *Shinya et al., 2006*; *Bouvier and Lowen, 2010*). Consequently, the ferret model has been used to assess numerous aspects of influenza transmission potential including: phenotypic traits associated with transmission (*Belser et al., 2008*, *2013*; *Song et al., 2009*; *van Doremalen et al., 2011*; *Blumenkrantz et al., 2013*), transmission under antiviral prophylaxis (*Oh et al., 2014*), and the relative transmissibility of drug resistant (*Herlocher et al., 2004*; *Hurt et al., 2010*; *Kiso et al., 2010*; *Seibert et al., 2010*; *Duan et al., 2011*; *Hamelin et al., 2011*), emerging (*Maines et al., 2006*; *Itoh et al., 2009*; *Belser et al., 2013*; *The SJCEIRS Working Group, 2013*; *Watanabe et al., 2013*; *Zhu et al., 2013*; *Xu et al., 2014*), or lab-created isolates (*Herfst et al., 2012*; *Imai et al., 2012*; *Sutton et al., 2014*).

Despite the widespread use of ferrets to assess transmission of influenza, the suitability of ferrets to assess pandemic potential in humans remains unknown, because the relationship between

**eLife digest** Every year, thousands of people develop influenza (flu). After being infected by the influenza virus, the immune systems of most people adapt to fight off the virus if it is encountered again. However, there are many different strains of influenza, and new strains constantly evolve. Therefore, although someone may have developed resistance to one previously encountered strain, they can still become ill if another strain infects them.

Different strains of the influenza virus have different abilities to spread between people and make them ill. One way that scientists assess whether a particular strain of influenza is a threat to people is by studying ferrets, which develop many of the same flu symptoms as humans. However, questions have been raised over how accurately ferret studies reflect whether a particular virus strain will spread between humans.

Controversy has also arisen over experiments in which ferrets are infected with genetically engineered strains of influenza that mimic how a strain that has evolved in birds could adapt to cause a pandemic in humans. In 2014, the United States government suggested that such research should be temporarily stopped until more is known about the risks and usefulness of these studies. Now, Buhnerkempe, Gostic et al. have compared the results of 240 ferret and human studies that aimed to assess how easily strains of influenza spread. Specifically, the studies looked at how often a healthy ferret or human became ill when exposed to an animal or human infected with a particular strain of influenza.

The results of the ferret transmission studies matched well with transmission patterns observed in human studies. Ferret studies that assessed how the influenza virus is transmitted through the air via sneezes and coughs were particularly good at predicting how the virus spreads in humans. But Buhnerkempe, Gostic et al. caution that ferret studies are not always accurate, partly because they involve small numbers of animals, which can skew the results. There also needs to be more effort to standardize the procedures and measurements used in ferret studies.

Still, the analysis suggests that overall, ferret studies are a useful tool for making an initial prediction of which influenza strains may cause a pandemic in humans, which can then be verified using other methods.

transmission in ferrets and in humans has never been assessed quantitatively (*Palese and Wang, 2012*; *Casadevall and Imperiale, 2014*; *Lipsitch, 2014*). In fact, conspicuous differences in ferret and human transmissibility for influenza A(H7N9) have cast doubt on the validity of the ferret model for assessing transmission in humans (*Lipsitch, 2013*). As a consequence, ferret studies can only be interpreted, strictly, in terms of general mammalian transmissibility (*Herfst et al., 2012*; *Imai et al., 2012*; *Casadevall and Imperiale, 2014*; *Casadevall et al., 2014*).

Furthermore, the recent controversy surrounding 'gain-of-function' (GOF) experiments on highly pathogenic avian influenza A(H5N1) in ferrets (*Herfst et al., 2012*; *Imai et al., 2012*) and proposed GOF experiments on A(H7N9) viruses (*Fouchier et al., 2013*) has led to ethical questions about influenza GOF experiments and scientific questions about the use of ferrets to assess transmission (*Morens et al., 2012*; *Casadevall and Imperiale, 2014*; *Casadevall et al., 2014*; *Lipsitch, 2014*; *Lipsitch and Galvani, 2014*; *Russell et al., 2014*). With the U.S. government halting funding and calling for a voluntary moratorium and period of review on such experiments as of October 2014 (*White House Office of Science and Technology Policy, 2014*), groups on all sides of the debate have issued renewed calls for studies on the link between influenza transmissibility in ferrets and in humans (*Morens et al., 2012*; *Lipsitch, 2013*, *2014*; *Casadevall and Imperiale, 2014*). Here, we address this gap by compiling ferret transmission studies and comparing their results to estimates of influenza transmission in humans.

## Results

### Comparing ferret and human secondary attack rates

To assess the quantitative relationship between influenza transmission in ferrets and in humans, we assembled data from all published ferret transmission studies that met our inclusion criteria, including

ferret experiments designed to test transmission in the presence of direct contact (co-housing) or by respiratory droplets (adjacent housing allowing air exchange). For each experiment, we calculated the secondary attack rate (SAR), which is defined as the probability of infection for a susceptible individual following known contact with an infectious individual (*Halloran, 2005*). To match the close contact found in ferret studies, we reviewed estimates of SAR in humans obtained from household contact data (*Figure 1*).

When comparing estimates of human and ferret SAR across subtypes, we found that, as expected (*Lakdawala and Subbarao, 2012*), ferret SAR estimates from current experimental designs do not quantitatively align with human SAR estimates—ferret SAR estimates are typically higher than the corresponding human estimate. However, ferret and human SAR estimates are correlated. For respiratory droplet experiments, the ordering of subtypes by ferret SAR was similar to that in human SAR (*Figure 1A,B*), and mean ferret respiratory droplet SAR explained 66% of the variation in mean human SAR estimates across subtypes (p = 0.003, *Figure 2A*). Direct contact transmission in ferrets was not significantly related to human SAR at the subtype level (p = 0.14, *Figure 2A*), suggesting that

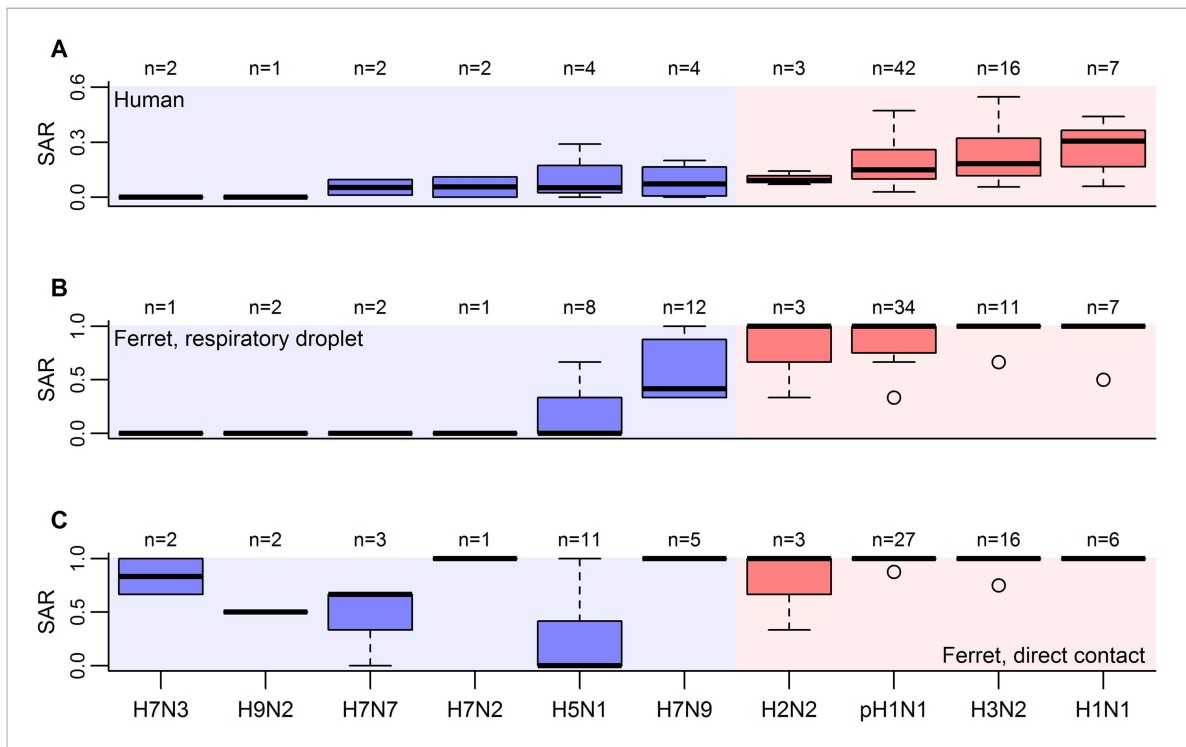

**Figure 1**. Boxplots of influenza SAR estimates by subtype. (**A**) Human SAR, (**B**) ferret respiratory droplet SAR, and (**C**) ferret direct contact SAR. Solid, black lines represent the subtype medians. Boxes give the inter-quartile range with whiskers extending out up to 1.5 times this range. Points represent extreme values. The number of estimated SARs for each subtype is given above each box-and-whisker plot (n). Subtypes were ordered according to the mean human SAR value in all panels. Shading depicts the known human transmission pattern of the subtypes (red—supercritical; blue—subcritical).

The following source data and figure supplement are available for figure 1:

**Source data 1**. Estimates of human household SAR.

**Source data 2**. Ferret influenza transmission studies via respiratory droplets using human isolates.

**Source data 3**. Ferret influenza transmission studies via direct contact using human isolates.

**Source data 4**. Ferret influenza transmission studies via respiratory droplets and direct contact using avian isolates.

**Figure supplement 1**. Analysis supporting inclusion of SAR estimates from isolates generating using reverse genetics.

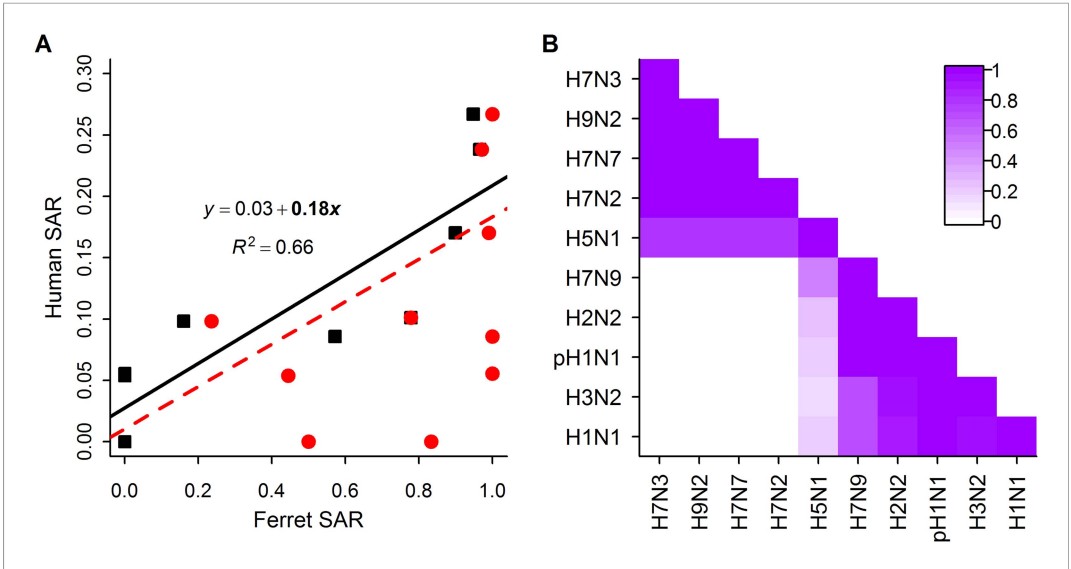

**Figure 2**. Analysis of subtype-specific SAR. (**A**) Comparison of human SAR and ferret SAR for ferret respiratory droplet (black squares) and direct contact (red circles). Data points are the mean human SAR by subtype vs the weighted mean ferret SAR by subtype, where weights are determined by the number of ferrets used in each experiment. Lines give the best fit weighted linear regression models with weights given by the number of human SAR estimates. The solid line indicates a significant relationship between ferret respiratory droplet SAR and human SAR described by the given equation (significant terms are bolded; p = 0.003), while the dashed line indicates a non-significant relationship (p = 0.14) for ferret direct contact transmission. (**B**) The degree of overlap in the distributions of ferret respiratory droplet SAR estimates for each subtype. Dark purple indicates subtypes with complete overlap, while white indicates no overlap.

The following figure supplement is available for figure 2:

**Figure supplement 1**. Analysis of subtype-specific SAR including avian isolates for H5N1 and H7N9.

for estimates of human-to-human transmissibility, direct contact experiments may have less value than respiratory droplet experiments.

Despite the strong relationship observed between mean ferret and human SAR estimates (*Figure 2A*), distributions of ferret SAR estimates for each subtype overlapped substantially (*Figure 2B*). These overlaps prevent the result from any given ferret experiment (e.g., on a novel, uncharacterized strain) from being unambiguously aligned with the transmission potential exhibited by any particular, previously-characterized subtype.

## Using ferret SAR to characterize human pandemic potential

To improve the power to assess pandemic potential, we specified two clusters of subtypes with distinct transmission patterns in humans: subtypes with sustained human-to-human transmission (i.e. supercritical; H1N1, H3N2, H2N2 and pH1N1) and subtypes without sustained human-to-human transmission (i.e. subcritical; H7N9, H5N1, H7N7, H7N2, H7N3 and H9N2). Using logistic regression, we identified ranges of ferret SAR that characterize supercritical and subcritical influenza viruses (*Figure 3*). Ferret respiratory droplet SAR was a significant predictor of the probability that a virus is supercritical or subcritical in humans (p < 0.0001; *Figure 3A*, *Table 1*). By accounting for the uncertainty in this relationship, we identified ranges of ferret SAR that indicate a high probability of strains being identified as supercritical or subcritical (*Figure 3A*). However, a range of intermediate ferret SAR values yielded equivocal results (i.e. the 95% confidence interval for classification included a classification probability of 0.5). Direct contact transmission was also a significant predictor of supercritical or subcritical transmission in humans (p = 0.01; *Figure 3B*, *Table 1*). Information theoretic model comparisons showed marginal support for a bivariate model using both respiratory droplet and

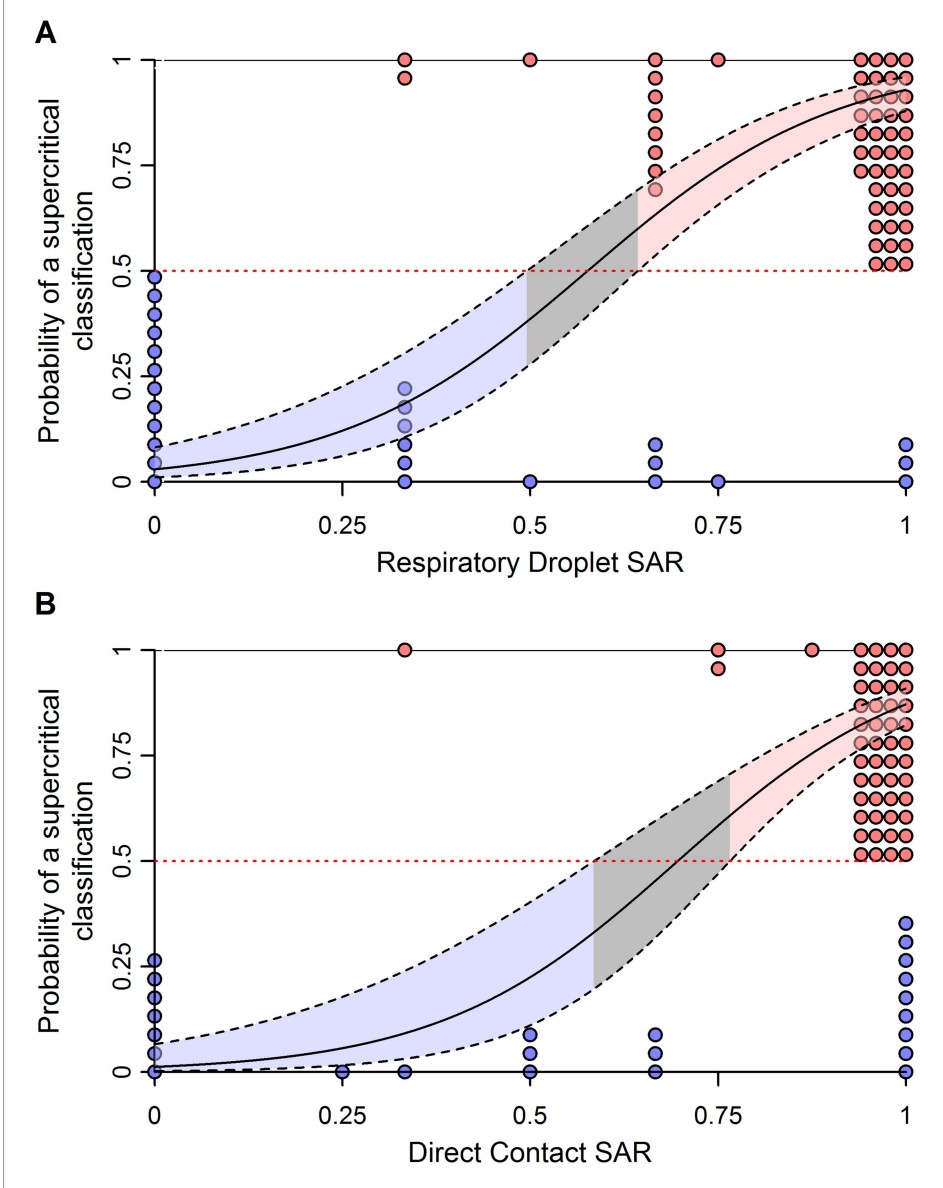

**Figure 3**. Weighted logistic regression predicting the probability of a supercritical classification based on ferret SAR. (**A**) Respiratory droplet SAR and (**B**) direct contact SAR. Solid black line gives the fit of the weighted logistic regression, where model weights are given by the number of ferrets in each experiment. Dashed black lines give the 95% confidence interval on the model predictions. Shading in the prediction interval represents values of SAR for which the 95% confidence intervals for predicted model fit do not overlap a probability of 0.5 (the dashed red line) indicating a high probability of being supercritical (red shading) or subcritical (blue shading). The gray shading represents SAR values where the 95% CI on the prediction overlaps 0.5, providing equivocal classification. Circles show the individual ferret SAR estimates (See *Figure 1—source data 2, 3*) for supercritical (top in red) and subcritical viruses (bottom in blue).

The following figure supplements are available for figure 3:

**Figure supplement 1**. Comparison of ferret SAR via respiratory droplet and direct contact transmission for single influenza isolates.

**Figure supplement 2**. Effect of uncertainty in ferret SAR on its relationship with the probability of being classified as supercritical.

*Figure 3. continued on next page*

*Figure 3. Continued*

**Figure supplement 3**. ROC curves for classifying pandemic potential using different definitions of transmission and transmission routes.

direct contact transmission data (*Table 1*). Considering the bivariate distribution of SAR estimates, however, it is clear that respiratory droplet SAR has the potential for greater specificity in predicting supercritical transmission (*Figure 3—figure supplement 1*).

The classification thresholds we identified for likely supercritical or subcritical subtypes account for uncertainties arising from the structure of our model, but not for uncertainties arising from the experimental data used to inform the model. Binomial uncertainties in ferret SAR data can be substantial, as ethical and logistic considerations limit sample sizes in these experiments (*Nishiura et al., 2013*). By re-fitting our logistic regression model to 1000 simulated datasets generated by binomial re-sampling of each data point, we found that the relationship between ferret SAR and a supercritical classification is quite robust to this uncertainty (*Figure 3—figure supplement 2*). However, while our analysis was fairly insensitive to binomial uncertainty within the aggregate data, attempts to classify SAR estimates from any individual experiment will be more sensitive to binomial uncertainty. For example, we applied our model to the most transmissible strains from two recent GOF studies on H5N1 avian influenza (*Imai et al., 2012*; *Herfst et al., 2012*; *Figure 4—source data 1*). All three strains had a ferret SAR that fell into the supercritical range, but the confidence intervals for the SAR estimates overlapped with the subcritical and/or equivocal ranges, preventing definitive classification (*Figure 4A*). Similarly, we found that studies on 1918 pandemic H1N1, a known pandemic strain, had ferret SAR estimates indicative of supercritical transmission, but again wide confidence intervals overlapped the subcritical and equivocal ranges (*Figure 4B*). SAR estimates for H7N9, known to be subcritical in humans, spanned the supercritical, subcritical, and equivocal ranges (*Figure 4C*). Even if results across all ferret respiratory droplet trials for H7N9 were aggregated into a single SAR estimate (representing 42 ferrets in all), we found an equivocal classification of human transmission pattern (*Figure 4C*). Consequently, care must be taken to avoid over-interpreting the results of ferret transmission studies.

## Discussion

For the first time, we have demonstrated a quantitative link between estimates of transmission efficiency of influenza among ferrets and among humans, at the subtype level. However, there is little power to resolve human SAR using ferret SAR estimates from single experiments. Instead, we observed ranges of ferret SAR distinguishing supercritical from subcritical subtypes that may be useful in identifying influenza viruses that pose greater or lesser risk of pandemic spread—especially for viruses with very high or low ferret SAR. In all analyses, including comparisons of sensitivity and false

**Table 1**. Parameter estimates for the weighted logistic regression relating human transmission class to ferret SAR

| Data | Model | $\beta_0$ | $\beta_{RD}$ | $\beta_{DC}$ | $\Delta$AIC |
|---|---|---|---|---|---|
| Full data | Direct contact | **−4.39** | - | **6.30** | - |
| | Respiratory droplet | **−3.52** | **6.10** | - | - |
| Restricted data | Respiratory droplet + direct contact | −1.76 | **8.72** | **−3.76** | 0 |
| | Respiratory droplet | **−3.77** | **6.42** | - | 3.623 |
| | Direct contact | **−3.07** | - | **3.74** | 57.348 |

Bolded estimates are significant at the $\alpha = 0.05$ level. Due to differing data between ferret respiratory droplet and direct contact transmission experiments, no model selection was done on the full data. Instead, model selection was done only for studies where authors performed respiratory droplet and direct contact transmission experiments on the same isolate.

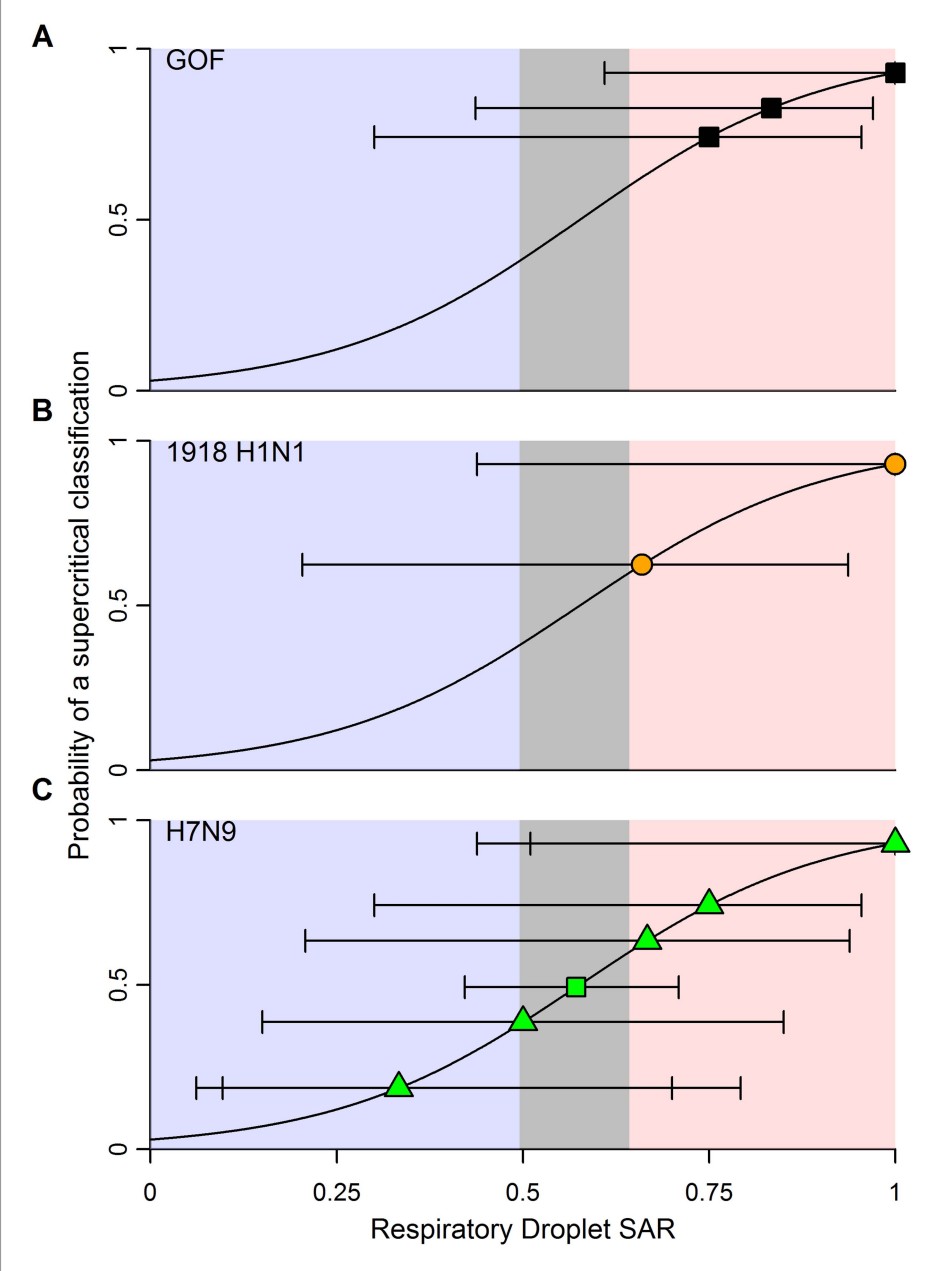

**Figure 4**. Predictions of the transmission pattern for current and historical isolates of concern. (**A**) Gain-of-function experiments with H5N1 avian influenza (*Herfst et al., 2012*; *Imai et al., 2012*), (**B**) the reconstructed 1918 pandemic H1N1 strain (*Tumpey et al., 2007*; *Imai et al., 2012*), and (**C**) H7N9 avian influenza. Solid black curves and shading represent the logistic regression fit and likely transmission pattern, respectively, as depicted in *Figure 2*. Horizontal lines give the 95% Wilson-score interval for each binomial estimate. In all panels, transmission is defined using seroconversion and viral isolation in nasal washes. In (**C**), green triangles represent individual experiments, while the green square is the aggregated data across all twelve H7N9 transmission experiments in ferrets. Notice that 6 data points are represented at a SAR of 0.33 and 3 at a SAR of 1. See *Figure 1—source data 2* and *Figure 4—source data 1* for full data.

The following source data and figure supplement are available for figure 4:

**Source data 1**. Ferret influenza transmission studies via respiratory droplets using strains from gain-of-function experiments with H5N1 avian influenza and the reconstructed 1918 pandemic H1N1 strain.

**Figure supplement 1**. Sample size calculations.

positive rate of various classification thresholds (*Figure 3—figure supplement 3*), we found that respiratory droplet transmission in ferrets was a better indicator of transmission in humans than direct contact transmission. However, direct contact experiments used in conjunction with respiratory droplet experiments can provide additional information on transmission in humans.

Sample size is a serious challenge to operational use of the results shown here. The largest sample size we found in our review of transmission studies was twelve ferrets (*Herlocher et al., 2002*). Even for a supercritical strain with an assumed ferret SAR of 1, 8 ferrets must be tested to classify that strain as supercritical with 80% power at a significance level of 0.05 (*Figure 4—figure supplement 1*). For an assumed ferret SAR of 0.8—more in line with zoonotic strains of interest (*Figure 4*), but closer to the lower end of the supercritical range—achieving the same power would require more than 30 ferrets (*Figure 4—figure supplement 1*). Such a sample size is obviously prohibitive. It is important to note, though, that data from future experiments should refine the relationship in *Figure 3*, expanding the ranges corresponding to subcritical and supercritical transmission, and hence lowering the sample size requirements somewhat.

Other design changes could also enhance the value of ferret transmission experiments for informing risk assessments. In particular, it is vital to standardize experimental design in order to reduce noise and strengthen inference, beginning with establishing standard definitions of transmission for ferret experiments (i.e. viral titers in nasal washes vs serologic evidence). Discord between viral isolation and antibody data within a single experiment highlights this need and shows that serologic data is often a more sensitive metric of pathogen exposure (*Figure 1—source data 2, 3*). It has been questioned whether seroconversion always reflects a productive viral infection, but recent imaging studies indicate that seroconversion can detect infections that manifest deep in the respiratory tract, which would be missed by nasal wash measurements (*Karlsson et al., 2015*). Although all of our results were robust to the choice of transmission definition ('Results' not shown), defining transmission by viral isolation alone slightly increased predictive power for direct contact experiments, and slightly decreased predictive power for respiratory droplet experiments (*Figure 3—figure supplement 3*). Ultimately, this suggests that transmission should be assessed using both serological and viral data to aid in comparisons across experiment types, while allowing for exploration of exposure vs active infection. Dosing protocols can also vary widely across and within studies, in terms of viral titer and volume and even incompatible units. Standardized dosing protocols could reduce variability in ferret SAR estimates substantially. Additional data on time to infection, clinical signs, and mechanistic insights such as receptor binding affinities, none of which are systematically collected under standard protocols, could add value to ferret studies by giving additional power to differentiate among influenza viruses and subtypes with similar transmission outcomes.

Despite these challenges, ferret transmission experiments can contribute distinctive insights into the pandemic potential of novel influenza isolates. Our results show that ferret experiments provide a tool with relatively high sensitivity and specificity for identifying strains that may be supercritical in humans (*Figure 3—figure supplement 3*). Based on current scientific knowledge, risk screening might also incorporate high-throughput virologic and genetic screens used to identify isolates of concern by looking for genetic changes associated with altered binding affinities and other markers of transmission in mammals (*Russell et al., 2012*). Ultimately, however, human transmission is a complex and partially understood phenotype that is difficult to predict using these initial screens (*Russell et al., 2014*). Ferrets can provide a potential link between underlying virologic and genetic changes and potential transmissibility in humans. Future analyses should attempt to simultaneously incorporate data on the presence of specific mutations (e.g., PB2-K627E, *Van Hoeven et al., 2009*) and virologic factors (e.g., binding to α2-6 sialic acid glycans, *Belser et al., 2008*) into the present analysis of ferret transmissibility to determine if these genetic and virologic screens provide additional information on human transmission not captured by ferrets alone.

The resolution of our analysis was limited to the subtype level, because human transmissibility data are not available for more specific strains. Some caution is needed when assessing transmission of novel isolates within a subtype, as out-of-sample predictions under this scenario are likely to be particularly hazardous. As a first assessment of the effect of within-subtype variation, we re-ran our analyses in *Figure 2* for a broader dataset including H5N1 and H7N9 strains isolated from avian hosts (*Figure 1—source data 4*; *Figure 2—figure supplement 1*). The results were consistent with our main findings, giving some confidence that our results are robust to such within-subtype variation (*Figure 1—source data 4*; *Figure 2—figure supplement 1*). Additional data on consensus viral

sequences within human outbreaks are needed to relate human SAR estimates more specifically to isolates tested in ferrets and clarify the effect of within-subtype variation on predicted human transmission behavior. In the absence of these data, our analysis represents a new null model against which deviations within subtypes can be measured to identify strains that can provide additional information on the molecular features associated with transmissible phenotypes in ferrets and/or humans.

Recently, the obvious disparity between highly efficient H7N9 transmission in some ferret experiments and inefficient H7N9 transmission in humans (see *Figure 2B*, *Figure 4C*) led to questions about the general validity of the ferret transmission model (*Lipsitch, 2013*). Our results at least partially assuage these concerns. In spite of the substantial variation we observed within H5N1 and H7N9 subtypes, our results show that, statistically, isolates more transmissible in ferrets are more likely to be capable of sustained transmission in humans. Yet our data also demonstrate that the ferret transmission model is fallible: for H7N9, an emerging virus of great concern, ferret transmission experiments sometimes yield results that obviously contradict observed patterns in humans. These results are anomalous within the general mapping of ferret transmissibility to human transmissibility and thus, as mentioned previously, may present an opportunity to gain new insight into the molecular drivers of this complex phenotype. However, when screening emerging influenza viruses for pandemic potential, both false negatives and false positives have important consequences for health policy decisions. The deviations of H7N9 from the general correlation between human and ferret transmissibility underscore the importance of corroborating transmission estimates from the ferret model with other lines of evidence. The ultimate evidence to corroborate human transmission comes from epidemiological patterns of infection in humans. For a true pandemic influenza virus, however, such data are likely to come too late, highlighting the need for reliable methods to provide early warning on strains with pandemic potential.

Here we have put forward the first guidelines for translating the results of ferret experiments into a measure of pandemic potential in humans. Given the continued use of ferrets in other areas of influenza research (e.g., vaccine development), this finding enhances the broad value of ferret experiments. However, given pragmatic limitations on sample sizes in ferret studies, uncertainties in ferret SAR estimates are likely to limit the operational utility of these guidelines. This coupled with the biological complexities underlying transmissibility suggests that, at this time, ferret transmission data provide a valuable but imperfect correlate of human transmissibility, and further evidence is needed to assess whether other lines of evidence can improve this predictive capacity.

## Materials and methods

### Secondary attack rates

Most ferret transmission studies report the number of secondary infections amongst a specified number of naïve ferrets that are exposed to single inoculated individuals. This enables calculation of the SAR, which is the probability of infection for a susceptible individual following a known contact with an infectious individual (*Halloran, 2005*) and establishes a metric of transmissibility in ferrets that is directly comparable to household SAR in humans.

We obtained estimates of SAR in humans from household contact data using two methods. Ad hoc SAR estimates are obtained by taking the ratio of infected household contacts over total household contacts. This method is widely used, but may overestimate SAR, as it assumes each household experiences only one disease introduction (the index case) and ignores the possibility of multiple household exposures to an exogenous reservoir (*Longini et al., 1982*). Meanwhile, maximum likelihood procedures for SAR estimation use statistical models to simultaneously estimate the probability of secondary transmission within a household (SAR) and the probability of infection from the community (or other source). Thus, these estimates attempt to correct for the possibility of multiple introductions from an exogenous source (*Longini and Koopman, 1982*; *Longini et al., 1982*). However, even these estimates can be strongly skewed by the inclusion or exclusion of specific clusters, especially early in an outbreak when data is limited (*Aditama et al., 2012*). Furthermore, variation in existing, population-level immunity to specific strains, and the use of different case ascertainment methods in specific studies also inevitably skew estimates made using either procedure. Because each method has unique biases and limitations, we used published estimates of SAR based on either method, or calculated an ad hoc SAR estimate ourselves from data on the total and infected

number of household contacts in an outbreak. Human SAR estimates are only considered in our initial regression analysis (*Figure 2A*), so they do not influence our classification model (*Figure 3*).

## Literature review

To assess the relationship between human and ferret transmissibility of influenza, we reviewed existing estimates of subtype-specific SAR in humans and ferrets. We searched PubMed and Web Of Science [v5.15] databases using the following queries: (influenza AND household AND transmission AND H#N#) and (influenza AND 'secondary attack rate' OR SAR AND human AND H#N#) for human studies and (influenza AND transmission AND ferret* AND H#N#) for ferret studies. We repeated searches for subtypes H1N1, H7N9, H3N2, H7N7, H7N9, H7N2, H9N2, H5N1, H7N3, and H2N2. To ensure comprehensive coverage, additional studies were identified using reference lists from search results and additional spot searches were also conducted. We excluded isolates that represented outliers from identified subtypes (i.e. 1918 pandemic H1N1 (*Tumpey et al., 2007*) and novel swine-origin H3N2 in 2009–10 (*Pearce et al., 2012*)). Searches were completed on 20 July 2015.

Although the transmission potential of unique isolates within a subtype may vary, SAR in humans was reported only at the subtype level, preventing us from analyzing isolate-specific transmission potential. Overall, we found data for all three measures (ferret direct contact, ferret respiratory droplet, and human SARs) for ten influenza A subtypes: H7N3, H9N2, H7N7, H7N2, H5N1, H7N9, H2N2, pH1N1 (i.e. influenza A(H1N1)pdm09 virus), H3N2, and seasonal H1N1 (*Figure 1*).

## Inclusion criteria for ferret studies

We excluded ferret transmission studies that included serial passage of human isolates in ferrets prior to transmission experiments. To maintain consistency in transmission mechanisms, we excluded studies that inoculated ferrets by routes other than intranasal with a liquid inoculum (e.g., ocular inoculation or aerosol inhalation) and that inoculated ferrets with a lower viral dose than was typical for ferret transmission studies (<$10^3$ 50% egg infectious dose [$EID_{50}$]). We excluded studies where naive ferrets were not exposed to inoculated ferrets at 1 day post-inoculation, as was standard, and studies where the duration of contact was restricted. We also excluded trials in which ferrets were vaccinated or administered antiviral drugs for treatment or prophylaxis. If transmission of more than one subtype and/or isolate was tested in a single study (using different sets of immunologically naive ferrets for each isolate), we treated each subtype/isolate-specific data point separately. However, for some analyses, we grouped data from isolates belonging to the same subtype—the one exception being separation of 2009 pandemic H1N1 isolates (pH1N1) and pre-2009 H1N1 isolates (H1N1).

We distinguished between direct contact transmission experiments (in which sentinel ferrets were co-housed with the donor ferret) and respiratory droplet transmission experiments (in which ferrets were housed in adjacent cages designed to allow for airborne exchange, but in which direct or indirect contact between sentinels and donors is not possible). Transmission amongst ferrets was determined in each study using either a viral titer in nasal washes or a positive serologic test (i.e. hemagglutination inhibition assay) or by a combination of both tests. We noted any discrepancies between the two transmission mechanisms (*Figure 1—source data 2*, *Figure 1—source data 3*) and conducted analyses that showed our results were relatively robust to the transmission definition used (*Figure 3—figure supplement 3*).

To promote quality of comparison between ferret and human studies, we only included data from ferret studies that tested one or more wild-type human isolates. While avian and other animal isolates maintain close sequence homology with human isolates (*Claas et al., 1998*), the transmission of animal isolates into humans is associated with genetic bottlenecks (*Zaraket et al., 2015*) and considerable within-host adaptation (*Linster et al., 2014*). These evolutionary barriers lead to avian precursors that have lower mortality in mice, less morbidity in ferrets, and lower viral titers in human epithelial cells (*Belser et al., 2013*; *Watanabe et al., 2014*; *Zaraket et al., 2015*). Thus, these cross-species and within-host barriers have the potential to obscure the relationship between transmission in ferrets and transmission in humans, and we excluded avian and other animal strains from the main analysis as a result. We did, however, compile a database of ferret transmission experiments using avian isolates from subtypes H5N1 and H7N9 (*Figure 1—source data 4*) to test the validity of this exclusion. Avian isolates in these subtypes have the benefit of contemporary sampling in both space and time with their human counterparts. Supplementary analyses including

these avian isolates showed that our results were robust to the exclusion of non-human isolates (*Figure 2—figure supplement 1*).

We also included wild-type isolates from humans generated using reverse genetics techniques. Although viral isolates rescued through reverse genetic techniques are often assumed to have lower transmissibility, analyses with and without these rescued isolates yielded indistinguishable results. Indeed, for the small number of isolates for which we could make direct comparisons, isolates generated using reverse genetics exhibited similar transmissibility to their wild-type counterparts (*Figure 1—figure supplement 1*). Thus, our data set contained a total of 81 respiratory droplet (*Figure 1B*; *Figure 1—source data 2*) and 76 direct contact transmission trials (*Figure 1C*; *Figure 1—source data 3*).

## Inclusion criteria for human studies

Because we considered only household SAR, we excluded studies with non-standard household definitions (e.g., dormitories, health care centers, summer camps), and studies where household contacts could not be distinguished from broader community contacts. We also excluded data from studies of zoonotic strains where prior contact with potential livestock or wildlife reservoirs was noted for multiple contacts, thus hindering the distinction between primary and secondary cases. In order to represent a broad range of human SAR estimates, we included both prospective and retrospective household studies that either provided an explicit SAR estimate or reported data sufficient to calculate a SAR. This yielded a total of 83 estimates of human SAR (*Figure 1A*; *Figure 1—source data 1*).

## Analysis

Quantitative comparison of SAR in ferrets and SAR in humans was performed using linear regression (*Figure 2A*). Because human SAR estimates are not typically made for individual isolates, the comparison was done at the subtype level using the mean value of all estimates belonging to a subtype. For ferret experiments, we used a weighted mean by subtype, where the weights were given by the number of ferrets used in each experiment; for human estimates, we used the simple mean by subtype. The potential uncertainty in subtype mean SAR was large, especially for human SAR, where several emerging subtypes (i.e. H7N3, H9N2, H7N7, and H7N2) only had one or two estimates (*Figure 1A*). To allow for this uncertainty, we used a weighted linear regression with model weights given by the number of human SAR estimates.

To create *Figure 2B*, we developed an empirical measure for the overlap between distributions of ferret SAR estimates for pairs of subtypes that was a simple variant of other overlap indices used in ecology (*Ricklefs and Lau, 1980*). This was calculated by comparing the more transmissible and less transmissible of each of the subtypes, taking the minimum SAR estimate for the more transmissible subtype and the maximum SAR estimate for the less transmissible subtype, and counting the number of estimates for both subtypes that fell within this range of overlap (normalized by the total number of estimates for both subtypes). This yielded a measure between 0 and 1, where zero indicated that the ranges of observed SAR estimates for two subtypes were completely distinct and one indicated that the ranges completely overlapped, rendering the subtypes indistinguishable on the basis of SAR.

Examination of *Figure 2B* revealed two distinct clusters of subtypes whose distributions of SAR estimates overlapped almost completely. H1N1, H3N2, pH1N1, and H2N2 are supercritical subtypes with sustained transmission among humans; H7N2, H7N7, H9N2, and H7N3 are subcritical subtypes with weak transmission among humans. This grouping suggests there may be potential to use ferret SAR estimates for broader functional classification of viruses with or without pandemic potential. However, two subtypes of concern, H7N9 and H5N1 (both known to be subcritical in humans), were anomalies within the natural clusters we observed in *Figure 2B*: H7N9 clustered with supercritical subtypes, while H5N1 was weakly associated with both groups. We interpreted this as important biological variation within the group of subcritical subtypes, but considering the overarching interest in predicting whether particular subtypes might have pandemic potential, for our further analyses we chose to group subtypes according to their observed transmission pattern in humans (i.e. supercritical—H1N1, H3N2, pH1N1, H2N2 and subcritical—H7N3, H9N2, H7N7, H7N2, H5N1, H7N9).

To determine how ferret SAR was related to supercritical and subcritical classifications, we used a weighted logistic regression (*Figure 3*). Here, model weights are based on the number of ferrets used in each experiment, thus allowing for more confidence in estimates with larger numbers of

ferrets. Ferret SAR estimates that corresponded to a high probability of an isolate being classified as supercritical or subcritical (i.e. low probability of being supercritical) were determined by calculating 95% confidence intervals for the predicted model fit and identifying the ranges where these confidence intervals were either wholly above (supercritical) or wholly below (subcritical) a value of 0.5 (representing a random guess of supercritical or not). The sensitivity and specificity of various thresholds in ferret SAR were also assessed (*Figure 3—figure supplement 3*).

All analyses were done using R Statistical Software version 3.1.2 (*R Development Core Team, 2014*).

## Acknowledgements

MGB and JOL-S are supported by the Research and Policy for Infectious Disease Dynamics (RAPIDD) program of the Science and Technology Directorate, Department of Homeland Security, and Fogarty International Center, National Institutes of Health. KG is supported by National Institutes of Health under the Ruth L Kirschstein National Research Service Award (T32-GM008185). JOL-S is supported by the National Science Foundation (EF-0928690) and the De Logi Chair in Biological Sciences. MP is supported by the National Science Foundation Graduate Research Fellowship under Grant No. (DGE-1144087). Any opinion, findings, and conclusions or recommendations expressed in this material are those of the authors(s) and do not necessarily reflect the views of the National Science Foundation, the National Institutes of Health or the Centers for Disease Control and Prevention.

## Additional information

### Funding

| Funder | Grant reference | Author |
| --- | --- | --- |
| National Institutes of Health (NIH)/Fogarty International Center (FIC) | Research and Policy for Infectious Disease Dynamics (RAPIDD) | Michael G Buhnerkempe, James O Lloyd-Smith |
| National Institutes of Health (NIH) | Ruth L. Kirschstein National Research Service Award (T32-GM008185) | Katelyn Gostic |
| National Science Foundation (NSF) | DGE-1144087 | Miran Park |
| National Science Foundation (NSF) | EF-0928690 | James O Lloyd-Smith |
| University of California, Los Angeles (UCLA) | De Logi Chair in Biological Sciences | James O Lloyd-Smith |

The funders had no role in study design, data collection and interpretation, or the decision to submit the work for publication.

### Author contributions

MGB, KG, MP, Conception and design, Acquisition of data, Analysis and interpretation of data, Drafting or revising the article; PA, Conception and design, Acquisition of data, Drafting or revising the article; JAB, Acquisition of data, Analysis and interpretation of data, Drafting or revising the article; JOL-S, Conception and design, Analysis and interpretation of data, Drafting or revising the article

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
