## [Decision Letter]

Thank you for submitting your work entitled “Mapping influenza transmission in the ferret model to transmission in humans” for peer review at *eLife*. Your submission has been favorably evaluated by Prabhat Jha (Senior Editor), a Reviewing Editor, and two reviewers.

The following individuals responsible for the peer review of your submission have agreed to reveal their identity: Mark Jit (Reviewing Editor) and Marc Lipsitch (peer reviewer).

The Reviewing Editor and the reviewers discussed their comments before we concluded that your manuscript is a topical and valuable contribution that brings clarity to an area that has been in the realm of opinion and aggregated anecdotes. It is an important step forwarding in bridging mammalian transmission data with human epidemiological findings, and we commend you for attempting to tackle this imposing but important topic.

However, we have important reservations about it, particularly around the literature search, source data, its analysis and the description of the analysis in the text. In particular, there are major omissions in the source data tables which underpin the entire analysis. We strongly recommend (but do not insist) that you include as a contributing author a subject matter expert familiar with the breadth and scope of ferret transmission data, which may not be captured by a simple algorithmic PubMed search.

The following comments need to be addressed in a revised submission before it can be considered for publication:

1) More detail needs to be provided about the selection criteria for the analysis conducted in this assessment. While the manuscript states in the Methods that you “excluded subtypes for which there were no estimates of human or ferret SAR” (in the subsection “Literature Review”) and that you “excluded isolates that represented outliers from identified subtypes”, it is unclear to us how published H2N2 ferret studies do not meet the definition of providing estimations of “ferret SAR” or the selection criteria employed to determine which viruses constitute “outliers from identified subtypes”.

Furthermore, while you state (in the subsection “Inclusion criteria for ferret studies”) that “to promote quality of comparison between ferret and human studies, we included data only from ferret studies that tested one or more wild-type human isolates”, this exclusion removes numerous studies with avian viruses which nonetheless share close sequence homology with viruses isolated from humans; please provide additional text justifying this decision.

2) In Figure 1, please better define *“n”*. Does it reflect the number of individual ferret inoculated-contact pairs included in each ferret study? You state that “the number of estimates for each subtype is given above each box-and-whiskers plot” (Figure 1 legend) yet this does not clarify the true number of ferret inoculated-contact pairs represented in this graph. Do you give more weight to transmission studies which employ more animals or are all transmission studies weighed equally, even if one study uses an *n*=2 and another uses an *n*=4-6? This level of simplicity does not accurately reflect the complexity of data in the published literature and must be augmented and/or better described for the presented results if you intend for this data to be meaningful to researchers who work with ferret transmission models.

3) Upon first glance at [Supplementary-material SD1-data SD2-data SD3-data], it appears that there are missing studies and inaccuracies in the studies listed which you used to conduct their analyses. For example, in [Supplementary-material SD2-data], you list respiratory droplet H5N1 studies from 4 published studies, yet exclude ferret transmission data which appears to meet the criteria outlined which appears in studies by Jackson et al. (wild-type Thai/16 virus, PMID 19493997) and Herfst et al. (wild-type Indo/05 virus, PMID 22723413). Also in [Supplementary-material SD2-data], you appear to exclude ferret transmission data for respiratory droplet pH1N1 which meets the criteria outlined in the study by Lakdawala et al. (California/4/09, PMID 22241979). Furthermore, in [Supplementary-material SD2-data] and Figure 1, you list duplicate entries for respiratory droplet transmission of the H7N7 virus A/NL/230/03, yet the publication cited states that only direct contact transmission, and not respiratory droplet transmission, was performed in duplicate for this virus, meaning that there should be an *n*=2 for this virus subtype for RD transmission per your definitions, not an *n*=3. Did you consult an outside source (ideally a subject matter expert in ferret transmissibility studies) to verify/confirm their literature search? These are only some of the errors the reviewers identified in studies that do not appear to have been captured in this literature search.

4) The grouping of viruses by surface glycoprotein HA/NA subtype only seems highly simplistic, as this does not account for differences in species origins (i.e. swine vs. human vs. canine lineage H3N2 viruses) or within virus subtypes (i.e. North American vs. Eurasian lineage H7 viruses). This grouping also does not take into account potential differences in receptor binding specificity or the presence of particular mutations/amino acids throughout the virus which might contribute to increased/decreased virus transmissibility by either direct contact or respiratory droplets. You must include in the text a greater emphasis on how the limitations this simplistic grouping of viruses by surface glycoproteins only may adversely affect or otherwise influence your findings – the statement in the Discussion that “none of [these features] are systematically collected under standard protocols” is not sufficient. While it is understandable that some generalizations will need to be made when grouping viruses by subtype only, you should disclose in the text to more detail the potential caveats of such assumptions.

5) You rightfully mention in the third paragraph of the Discussion the need for a standard definition of transmission for ferret experiments (i.e. viral isolation vs seroconversion only vs. both), yet appear to use all definitions interchangeably in your analysis. From the perspective of researchers who perform ferret transmission studies, there is a large difference in transmission robustness between these two criteria, with the presence of both virus isolation and seroconversion considered the best evidence of successful virus transmission. This has notable bearing on the direct contact transmissibility analysis of this study, as viruses which transmit poorly by respiratory droplets are often able to lead to low-level seroconversion of naïve ferrets when placed in direct contact. If you are counting these low-level seroconversions as true virus transmission events, it is not surprising that the data regarding SAR from direct contact transmission exhibits greater variability and less statistical weight. You should disclose more fully this potential caveat in your analysis and interpretation of direct contact ferret transmission SAR.

6) The ferret data are treated in the statistical analyses as being without error, although as you acknowledge they are extremely noisy because of small sample size. Horizontal error bars are shown in Figure 4 but not in Figure 3. In the linear regression context there is something called Type 2 regression to handle this. I suspect there is something comparable in logistic regression, and it would be good to try it or at least point to it.

7) The conclusion at the end of the Discussion and throughout is not quite justified. Why should ferrets be a first-like screening tool, when other virologic studies are cheaper and higher-throughput? In practice ferret studies are done on a pretty restricted subset of isolates thought to be high-risk for other reasons, including epidemiological data and virologic findings. So the questions that arise are: conditional on being considered high-enough risk to merit ferret studies, what additional value do the ferret studies add? Should we really be doing ferret studies first? If so, what should be the means of follow-up?

In essence, the paper shows that there is a good correlation between ferret outcomes and human data. This is very useful. It also quantifies the noise and notes the sources of that noise. That is also very useful. We are not sure that it provides evidence in the current form of the argument for how ferrets can add value to pandemic prediction.

This leads to a broader point that is not much addressed. Pandemic risk prediction is used for the targeting of countermeasures to contain certain strains considered high risk. Thus having a good correlation may help to guide such efforts to on average more threatening strains, but may nonetheless produce many false negatives (failing to spot highly risky strains) and false positives (identifying strains as high risk that are not actually high risk). The rather bland conclusion that ferrets should keep being used seems to skirt this issue. We would like you either to modify your conclusions to present this point, or else to give stronger justification for the use of the ferret model to guide pandemic prediction.

8) In the subsection “Inclusion criteria for ferret studies”, the inclusion of only human isolates studied in ferrets should be directly highlighted. The data on H7N9 (and maybe others too) show that taking a strain from a bird into a ferret may give very different results (no transmission in ferrets) from the same subtype isolated from a human (transmission in ferrets). This has several implications. First, not everyone will note the subtlety but it probably affects the results of the study (it might have looked different if bird isolates tested in ferrets had been included). Second, it highlights the point that subtype is not an ideal level of aggregation, though it is needed for the reasons you discuss. Third, it emphasizes the ambiguity of the assessment of pandemic potential. If a strain that is nontransmissible in ferrets can change in possibly a single human passage into one that is, which readout in the ferrets gives the better estimate of pandemic potential?

[Editors' note: further revisions were requested prior to acceptance, as described below.]

Thank you for resubmitting your work entitled “Mapping influenza transmission in the ferret model to transmission in humans” for further consideration at *eLife*. Your revised article has been favorably evaluated by Prabhat Jha (Senior Editor), Mark Jit (Reviewing Editor) and Marc Lipsitch (peer reviewer).

The manuscript has been improved but there are some remaining issues that need to be addressed before acceptance, as outlined below:

1) In the fourth paragraph of the Discussion, you advocate screening strains for virologic and genetic markers of pandemic potential before selecting a subset (presumably positive on all these screens) for ferret testing. There is no evidence that this is a good idea. To demonstrate that, a larger analysis would need to be done on at least the included datasets here (but really one would want to know about strains on which ferret studies were not done, too) to see if that screen would have been a good idea. The fact that one of the most “popular” mutations was not present in H1N1p (PB2 E627K) casts doubt on this strategy. We would suggest you modify this paragraph to speculate that further studies (perhaps replicating their approaches) would be needed to clarify whether virologic and genetic data add anything to ferret findings.

2) Again in the Discussion, you state: “The ultimate evidence to corroborate human transmission comes from epidemiological patterns of infection in humans, but for a true pandemic influenza, this data may come too late highlighting the importance of ferrets in early identification of these strains”. This conclusion seems to beg the question and not to follow from the rest of the paragraph. You have clearly documented that while very often informative, the results of ferret experiments can sometimes be misleading. Human transmission data highlights the *need* for a reliable early warning system that *can reliably identify* strains of pandemic potential, but not the “*importance* of ferrets in the early identification of these strains.” Ferrets can contribute, but a reliable early warning system is still needed.

3) Also in the Discussion, you state: “This coupled with the biological complexities underlying transmissibility suggests that, at this time, ferret transmission data are a valid tool for assessing human transmissibility, but should be corroborated by further virologic and epidemiologic research”. We do not have any valid tool for assessing transmissibility, though we do have useful correlates that improve our estimate. And saying we need further virologic and epidemiologic evidence suggests that this would be dispositive, but we don't have virologic predictors that are reliable, and epidemiologic evidence comes too late as you point out. It may be better to write something like “[…] at this time, ferret transmission data provide a valuable but imperfect correlate of human transmissibility, and further evidence is needed to assess whether other lines of evidence can improve this predictive capacity.”

4) In the second paragraph of the Discussion you state: “Sample size is a serious challenge to operational use of the results shown here. The largest sample size we found in our review of transmission studies was twelve ferrets (17). Even if 8/8 contact ferrets were infected in an experiment, there is only 80% power at a significance level of 0.05 to classify that strain as supercritical (Figure 4—figure supplement 1).” This is difficult to understand because a power calculation can only be done before specifying the outcome of the binomial experiment, and not after. Something seems to be wrong with this.

---

## [Author Response]

1) More detail needs to be provided about the selection criteria for the analysis conducted in this assessment. While the manuscript states in the Methods that you “excluded subtypes for which there were no estimates of human or ferret SAR” (in the subsection “Literature Review”) and that you “excluded isolates that represented outliers from identified subtypes”, it is unclear to us how published H2N2 ferret studies do not meet the definition of providing estimations of “ferret SAR” or the selection criteria employed to determine which viruses constitute “outliers from identified subtypes”.

Furthermore, while you state (in the subsection “Inclusion criteria for ferret studies”) that “to promote quality of comparison between ferret and human studies, we included data only from ferret studies that tested one or more wild-type human isolates”, this exclusion removes numerous studies with avian viruses which nonetheless share close sequence homology with viruses isolated from humans; please provide additional text justifying this decision.

We added additional text justifying our exclusion of avian and other animal isolates (in the subsection “Inclusion criteria for ferret studies”). Briefly, despite sequence homology between avian and human isolates, the cross-species jump from birds to humans involves evolutionary barriers, including transmission bottlenecks and within-host adaptation. Considering only human isolates removes uncertainties generated by some of these barriers, potentially providing a clearer signal on the relationship between transmission in ferrets and transmission in humans. To test the validity of these omissions, we also added data on ferret experiments performed with H5N1 and H7N9 avian isolates. We chose isolates from these subtypes given their position at the boundary between sub- and supercritical subtypes (Figure 2) and the concurrence in both space and time between isolation in birds and people. However, inclusion of avian strains has little impact on the correspondence between ferret and human transmission (Figure 2 and Figure 2—figure supplement 1), suggesting that an extra review of animal isolates would not contribute added insight to this analysis.

Additionally, we added experiments using H2N2 and H7N3 isolates to the analysis after finding appropriate estimates of ferret and human SARs, respectively, in the literature.

*2) In*
Figure 1*, please better define “*n*”. Does it reflect the number of individual ferret inoculated-contact pairs included in each ferret study? You state that “the number of estimates for each subtype is given above each box-and-whiskers plot” (*Figure 1
*legend) yet this does not clarify the true number of ferret inoculated-contact pairs represented in this graph. Do you give more weight to transmission studies which employ more animals or are all transmission studies weighed equally, even if one study uses an* n*=2 and another uses an* n*=4-6? This level of simplicity does not accurately reflect the complexity of data in the published literature and must be augmented and/or better described for the presented results if you intend for this data to be meaningful to researchers who work with ferret transmission models.*

We have clarified that *“n”* is the number of estimated SARs for each subtype in the legend for Figure 1. Given that there is likely more variability between laboratories (e.g. dosing or cage setups) than between any two ferrets, the unit of this analysis is necessarily an experiment. We do agree with the reviewers that higher weight should be placed on experiments that use more ferrets. We have redone the analyses using weighted means of the ferret SARs in Figure 2 and including the number of ferrets tested as case weights in the logistic regression (Figure 3). A description of this is included in the Methods (in the subsection “Analysis”) and in the legends for Figures 2 and 3.

*3) Upon first glance at*
[Supplementary-material SD1-data SD2-data SD3-data]*, it appears that there are missing studies and inaccuracies in the studies listed which you used to conduct their analyses. For example, in*
[Supplementary-material SD2-data]*, you list respiratory droplet H5N1 studies from 4 published studies, yet exclude ferret transmission data which appears to meet the criteria outlined which appears in studies by Jackson et al. (wild-type Thai/16 virus, PMID 19493997) and Herfst et al. (wild-type Indo/05 virus, PMID 22723413). Also in*
[Supplementary-material SD2-data]*, you appear to exclude ferret transmission data for respiratory droplet pH1N1 which meets the criteria outlined in the study by Lakdawala et al. (California/4/09, PMID 22241979). Furthermore, in*
[Supplementary-material SD2-data]
*and*
Figure 1*, you list duplicate entries for respiratory droplet transmission of the H7N7 virus A/NL/230/03, yet the publication cited states that only direct contact transmission, and not respiratory droplet transmission, was performed in duplicate for this virus, meaning that there should be an* n*=2 for this virus subtype for RD transmission per your definitions, not an* n*=3. Did you consult an outside source (ideally a subject matter expert in ferret transmissibility studies) to verify/confirm their literature search? These are only some of the errors the reviewers identified in studies that do not appear to have been captured in this literature search.*

With the help of Dr. Jessica Belser, we have considerably expanded the scope of our literature review. Specifically, we added 23 more estimates of ferret respiratory droplet SAR, 27 more estimates of ferret direct contact SAR, and 7 more estimates of human SAR. These additions also broadened our analysis to include two more subtypes, H2N2 and H7N3. These additional data, along with our weighted regression scheme, have led to narrower confidence intervals in our main analysis.

4) The grouping of viruses by surface glycoprotein HA/NA subtype only seems highly simplistic, as this does not account for differences in species origins (i.e. swine vs. human vs. canine lineage H3N2 viruses) or within virus subtypes (i.e. North American vs. Eurasian lineage H7 viruses). This grouping also does not take into account potential differences in receptor binding specificity or the presence of particular mutations/amino acids throughout the virus which might contribute to increased/decreased virus transmissibility by either direct contact or respiratory droplets. You must include in the text a greater emphasis on how the limitations this simplistic grouping of viruses by surface glycoproteins only may adversely affect or otherwise influence your findings – the statement in the Discussion that “none of [these features] are systematically collected under standard protocols” is not sufficient. While it is understandable that some generalizations will need to be made when grouping viruses by subtype only, you should disclose in the text to more detail the potential caveats of such assumptions.

We agree that grouping at the subtype level may potentially obscure important complexities within select subtypes, including molecular determinants or other phylogenetic features which may contribute to virus transmissibility in mammalian models. However, this level of grouping was necessary to perform the correlations to human SAR and general humans transmission pattern (i.e. subcritical vs. supercritical) which represent the primary focus of this study. There simply are not sufficient data to estimate human transmissibility with resolution below the HA/NA subtype level. Additional text has been included in the Discussion to better clarify and justify this decision. Importantly, the grouping at subtype level employed here has provided a valid null model against which deviations can be identified, thus laying the foundation for potential further analyses investigating virus- or host-specific attributes which may contribute to these deviations, which we acknowledge in the fifth paragraph of the Discussion.

5) You rightfully mention in the third paragraph of the Discussion the need for a standard definition of transmission for ferret experiments (i.e. viral isolation vs seroconversion only vs. both), yet appear to use all definitions interchangeably in your analysis. From the perspective of researchers who perform ferret transmission studies, there is a large difference in transmission robustness between these two criteria, with the presence of both virus isolation and seroconversion considered the best evidence of successful virus transmission. This has notable bearing on the direct contact transmissibility analysis of this study, as viruses which transmit poorly by respiratory droplets are often able to lead to low-level seroconversion of naïve ferrets when placed in direct contact. If you are counting these low-level seroconversions as true virus transmission events, it is not surprising that the data regarding SAR from direct contact transmission exhibits greater variability and less statistical weight. You should disclose more fully this potential caveat in your analysis and interpretation of direct contact ferret transmission SAR.

We added additional data to [Supplementary-material SD2-data SD3-data] to address this, specifically assessing transmission by either virus isolation or seroconversion per the raw data provided in individual published studies, in lieu of relying on conclusions of the authors of the individual studies, which could vary between laboratories. Here, we specified experiments where results from viral isolation did not match seroconversion results. Our results were robust to the definition of transmission, with only the analysis of AUC showing any discernable impact of considering viral isolation alone (slightly positive impact for direct contact, negative for respiratory droplet). We added a recommendation that in practice both measures are needed to allow for comparisons across experiment types in the third paragraph of the Discussion.

*6) The ferret data are treated in the statistical analyses as being without error, although as you acknowledge they are extremely noisy because of small sample size. Horizontal error bars are shown in*
Figure 4
*but not in*
Figure 3*. In the linear regression context there is something called Type 2 regression to handle this. I suspect there is something comparable in logistic regression, and it would be good to try it or at least point to it.*

Because we are interested in predicting pandemic potential from ferret SAR and not in the regression coefficients themselves, model I regression (i.e. ordinary logistic regression) is the preferred statistical model as it minimizes squared residuals in the y direction, whereas model II regression approaches do not (Sokal and Rohlf, 1995, p. 545, Table 14.3; Legendre and Legendre, 1998). However, we agree with the reviewers that error in ferret SAR estimates will have an impact on setting super- and subcritical thresholds using 95% confidence bands on model predictions (Sokal and Rohlf, 1995, p. 545, Table 14.3; Legendre and Legendre 1998). To explore this uncertainty, we used a simulation approach. Here, we simulated 1000 datasets by taking binomial samples from each data point using a probability given by the observed ferret SAR and N equal to the number of ferrets used. To introduce uncertainty into those experiments where the ferret SAR was 0 or 1, we set the binomial probability to be 0.1 or 0.9, respectively. The results of this analysis are given in the subsection “Using ferret SAR to characterize human pandemic potential” and in Figure 3—figure supplement 2 and indicate that while uncertainty in the ferret SAR estimates impacts our results quantitatively, the underlying relationship between ferret SAR and an isolate’s general transmission behavior is robust to such uncertainty.

7) The conclusion at the end of the Discussion and throughout is not quite justified. Why should ferrets be a first-like screening tool, when other virologic studies are cheaper and higher-throughput? In practice ferret studies are done on a pretty restricted subset of isolates thought to be high-risk for other reasons, including epidemiological data and virologic findings. So the questions that arise are: conditional on being considered high-enough risk to merit ferret studies, what additional value do the ferret studies add? Should we really be doing ferret studies first? If so, what should be the means of follow-up?

It was not our intention to suggest that ferret studies should be the very first test done to identify strains of concern. Rather, we aimed to emphasize that they are not the final word. We have changed the language in the fourth paragraph of the Discussion to stress that, while virologic and genetic screening tools can indeed identify strains of concern, transmission is a complex phenotype that cannot be completely characterized using these high-throughput initial screens. Ultimately, transmission is a process occurring at the level of whole animals (and pairs of animals), and given our current state of knowledge, animal experiments continue to contribute distinct insights into transmission biology. Thus, ferrets provide a valuable missing link from virologic and genetic changes to a more complete understanding of their impact on human transmission behavior.

In essence, the paper shows that there is a good correlation between ferret outcomes and human data. This is very useful. It also quantifies the noise and notes the sources of that noise. That is also very useful. We are not sure that it provides evidence in the current form of the argument for how ferrets can add value to pandemic prediction.

This leads to a broader point that is not much addressed. Pandemic risk prediction is used for the targeting of countermeasures to contain certain strains considered high risk. Thus having a good correlation may help to guide such efforts to on average more threatening strains, but may nonetheless produce many false negatives (failing to spot highly risky strains) and false positives (identifying strains as high risk that are not actually high risk). The rather bland conclusion that ferrets should keep being used seems to skirt this issue. We would like you either to modify your conclusions to present this point, or else to give stronger justification for the use of the ferret model to guide pandemic prediction.

We agree that predictions from ferret studies have the potential to identify many false negatives and false positives, although our results do indicate that sensitivity and specificity are relatively high when using ferret SAR to identify supercritical strains. Given the complexities underlying transmission, it is likely that ferret transmission studies, as a holistic measure of transmission, provide higher sensitivity and specificity than initial screens. While the current analysis does not allow us to verify this fact, it does provide the opportunity for future analyses to include all screening tools in predictions of human pandemic potential, thus increasing sensitivity and specificity further. We have included a discussion of this point, and explicit reference to the hazards of false negatives and false positives, in the fourth and sixth paragraphs of the Discussion.

8) In the subsection “Inclusion criteria for ferret studies”, the inclusion of only human isolates studied in ferrets should be directly highlighted. The data on H7N9 (and maybe others too) show that taking a strain from a bird into a ferret may give very different results (no transmission in ferrets) from the same subtype isolated from a human (transmission in ferrets). This has several implications. First, not everyone will note the subtlety but it probably affects the results of the study (it might have looked different if bird isolates tested in ferrets had been included). Second, it highlights the point that subtype is not an ideal level of aggregation, though it is needed for the reasons you discuss. Third, it emphasizes the ambiguity of the assessment of pandemic potential. If a strain that is nontransmissible in ferrets can change in possibly a single human passage into one that is, which readout in the ferrets gives the better estimate of pandemic potential?

We have added further clarification to the Methods about the exclusion of avian and other animal isolates (in the subsection “Inclusion criteria for ferret studies”). We also tested the validity of this exclusion by compiling data on avian H5N1 and H7N9 isolates. The inclusion of these isolates did not impact our results, suggesting that further review of these isolates is not currently necessary. Additionally, this also suggests that within-subtype variation is not obscuring broad-scale patterns. However, we added a discussion of within-subtype variation to address the second and third points raised by the reviewers, and a clarification that despite the subtype aggregation, we view high ferret SARs as indicative of a high likelihood of human-human transmission, regardless of subtype (see the sixth paragraph of the Discussion). While we agree with the reviewers’ final point that evolutionary dynamics within and between hosts are a necessary part of predicting pandemic potential, and indeed this is an important and fast-moving scientific frontier, this is beyond the scope of this study. Gain-of-function studies and studies of natural isolates are all trying to get at this point, and ethical concerns aside, we have shown that the use of ferrets to draw comparisons to humans in these studies is scientifically justified.

[Editors' note: further revisions were requested prior to acceptance, as described below.]

1) In the fourth paragraph of the Discussion, you advocate screening strains for virologic and genetic markers of pandemic potential before selecting a subset (presumably positive on all these screens) for ferret testing. There is no evidence that this is a good idea. To demonstrate that, a larger analysis would need to be done on at least the included datasets here (but really one would want to know about strains on which ferret studies were not done, too) to see if that screen would have been a good idea. The fact that one of the most “popular” mutations was not present in H1N1p (PB2 E627K) casts doubt on this strategy. We would suggest you modify this paragraph to speculate that further studies (perhaps replicating their approaches) would be needed to clarify whether virologic and genetic data add anything to ferret findings.

We agree with the reviewers that this is an important avenue for future research. As suggested, we have modified this paragraph to stress that further studies are needed to integrate genetic data on specific mutations, such as PB2-K627E, and virologic changes, such as sialic acid binding affinity, into the current results to truly assess how high-throughput screens may add value to predictions of human transmissibility from ferret experiments alone.

*2) Again in the Discussion, you state: “The ultimate evidence to corroborate human transmission comes from epidemiological patterns of infection in humans, but for a true pandemic influenza, this data may come too late highlighting the importance of ferrets in early identification of these strains”. This conclusion seems to beg the question and not to follow from the rest of the paragraph. You have clearly documented that while very often informative, the results of ferret experiments can sometimes be misleading. Human transmission data highlights the* need *for a reliable early warning system that* can reliably identify *strains of pandemic potential, but not the “*importance *of ferrets in the early identification of these strains.” Ferrets can contribute, but a reliable early warning system is still needed.*

We agree the lack of human data highlights the need for a reliable early warning system and not the need for ferret studies specifically, especially given some of the issues we discuss. We have changed this sentence to stress that methods to provide reliable early warning on strains with pandemic potential are needed.

3) Also in the Discussion, you state: “This coupled with the biological complexities underlying transmissibility suggests that, at this time, ferret transmission data are a valid tool for assessing human transmissibility, but should be corroborated by further virologic and epidemiologic research”. We do not have any valid tool for assessing transmissibility, though we do have useful correlates that improve our estimate. And saying we need further virologic and epidemiologic evidence suggests that this would be dispositive, but we don't have virologic predictors that are reliable, and epidemiologic evidence comes too late as you point out. It may be better to write something like “[…] at this time, ferret transmission data provide a valuable but imperfect correlate of human transmissibility, and further evidence is needed to assess whether other lines of evidence can improve this predictive capacity.”

We agree that the suggested change is a valuable clarification and have changed the text accordingly.

*4) In the second paragraph of the Discussion you state “Sample size is a serious challenge to operational use of the results shown here. The largest sample size we found in our review of transmission studies was twelve ferrets (*[17]*). Even if 8/8 contact ferrets were infected in an experiment, there is only 80% power at a significance level of 0.05 to classify that strain as supercritical (*Figure 4—figure supplement 1*).” This is difficult to understand because a power calculation can only be done before specifying the outcome of the binomial experiment, and not after. Something seems to be wrong with this.*

We appreciate the reviewers pointing out this confusing wording. To remedy this, we have changed the wording to reflect that the power calculations are in fact a priori instead of post-hoc. We intend for this analysis to demonstrate the sample size necessary to detect a difference between an assumed ferret SAR of a future experiment and the lower threshold for a supercritical classification.